# Influence of Fineness Levels and Dosages of Light-Burned Dolomite on Portland Cement Performance

**DOI:** 10.3390/ma15165798

**Published:** 2022-08-22

**Authors:** Wenxiu Jiao, Aimin Sha, Zhuangzhuang Liu, Shuo Li

**Affiliations:** 1Key Laboratory for Special Area Highway Engineering of Ministry of Education, Chang’an University, Xi’an 710064, China; 2School of Highway, Chang’an University, Xi’an 710064, China

**Keywords:** light-burned dolomite powders, ordinary Portland cement, mechanical properties, hydration properties

## Abstract

The paper aims to understand the effect of light-burned dolomite powders (LBD) on ordinary Portland cement (OPC) and evaluate the influence of LBD dosages and fineness levels on the mechanical properties and hydration properties of OPC. The LBD/OPC pastes were prepared by OPC blended with LBD at various replacement dosages and fineness levels. The mechanical properties were studied by flexural and compressive strength tests, while the hydration properties were investigated by X-ray diffraction (XRD), thermogravimetric analysis (TG), differential scanning calorimetry (DSC), and reaction degree of LBD. Experiment results indicated that the flexural and compressive strength of LBD/OPC samples were higher than reference sample at all ages. The fineness levels of LBD was C (C-LBD) with 0.5–1.5 wt% dosages, and the fineness levels of LBD was B (B-LBD) with 1.5–2.5 wt% dosages can significantly improve the strength of cement-based materials. The main mineral components of LBD are MgO and CaCO_3_, of which MgO could react with water to form Mg(OH)_2_ quickly, and CaCO_3_ could hydrate with C_3_A to from hydrated calcium carboaluminate (C_3_A·CaCO_3_·11H_2_O), which prevents the conversion of AFt to AFm.

## 1. Introduction

Limestone filler can be utilized as a replacement of cement clinker to improve the particle composition and adjust and optimize the structure of the pores [1,2]. Besides, it has good economic and environmental benefits, such as reducing carbon dioxide emissions during cement production process, which has been accepted and applied in most countries of the world [3,4,5]. As the main associated mineral of limestone, dolomite is rich in dosage and widely distributed in the world [6,7]. In China, the Shanxi, Hubei, Guizhou, and Liaoning provinces and the coastal areas of Shandong are the main sources of dolomite. Use of dolomite as one of cement component has been the focus of many research efforts in recent years [8,9,10,11].

The main chemical components of dolomite are calcium carbonate and magnesium carbonate (CaCO_3_·MgCO_3_). In addition, dolomite is also the main mineral component of dolostone and dolomitic limestone [12,13]. Dolomite can be used in cement in two ways. First, as aggregates for the production of concrete, and second as a mineral admixtures of cement which plays the role of active gel additions [14]. The application of dolomite as a mineral admixture in cement can improve some properties of cement-based materials. For example, it can compensate for the volume shrinkage of concrete to a certain extent, and improve the flowability and working performance. There is no negative impact on the volume stability of cement-based materials [14,15]. However, when dolomite is used as mineral admixture, the strength of OPC will be reduced [8].

Calcium carbonate (CaCO_3_) and magnesium carbonate (MgCO_3_) are the main components of dolomite; the decomposition temperature of them is different and the difference is more than 200 °C [16,17]. By controlling the calcination temperature and calcination time of dolomite, the MgCO_3_ in dolomite can be fully decomposed without CaCO_3_ participating in the decomposition. Thereby, an LBD containing active magnesium oxide (MgO) and inert CaCO_3_ is obtained [17,18,19]. The decomposition process of dolomite was usually divided into two steps. Firstly, dolomite compound salt is decomposed to inert CaCO_3_ and MgCO_3_; secondly, MgCO_3_ is further decomposed into MgO.

Massive reaction mechanisms and performance analyses of dolomite in cement have been studied with the progressive development and maturation of cement admixtures. Xu et al. studied the reaction mechanism of dolomite powder in the Portland cement by XRD, TGA, MIP and SEM. Results have shown that with the increase of curing temperature, the chemical reactivity of dolomite was significantly higher than that of limestone. Therefore, the pore structure of Portland—dolomite cement was refined and its long-term compressive strength was improved [12]. Xu and Chen et al. investigated the hydration and mechanical properties of dolomite powder on calcium sulfoaluminate (CSA) cements. They found that the hydration of CSA cement was accelerated, and the strength loss of the CSA mortar was alleviated with the addition of dolomite powder [20]. Nguyen investigated the effect of dolomite powder as part of fine particles on the engineering properties of slag—cement based self-compacting mortar. The workability of self-compacting mortar was enhanced and the microstructure was improved as the addition of dolomite substituting slag powder [21]. The stability of hydrate phase combination in Portland cement containing dolomite after leaching, carbonization, and chloride exposure was studied by Machner. The hydrotalcite was formed in the dolomite cement paste, which can withstand high degrees of leaching and carbonation [22].

Although notable progress has been achieved in the research and application of dolomite in cement, few works in the literature have reported the study of LBD as mineral admixtures for cement concrete in previous research. The reasons are as follows: (i) There is insufficient research on whether LBD has similar physical and chemical effects to dolomite and limestone in cement-based materials; (ii) MgO as one of the main mineral components of LBD was able to participate in hydration reaction quickly, which is significantly different from the almost inert limestone powder; and (iii) the effects of LBD on the composition, microstructure and long term stability of the pastes are also unclear [23]. However, it should be noted that, as a relatively economical limestone associated mineral, LBD can react with water fast to form calcium hydroxide (Ca(OH)_2_), which can be used as an activator for cement-based materials. At the same time, a certain amount of LBD is beneficial to the development of pastes strength.

In order to clarify the mechanical properties of LBD in cement-based materials, and quantitatively analyze its active effect, this paper studies the influence of fineness levels, dosages and curing times of LBD/OPC pastes, respectively. The mechanical properties were studied by compressive and flexural strength tests, meanwhile the hydration properties were researched by XRD, TG, DSC and reaction degree of LBD. It is expected to provide a basis for the safety and efficient application of LBD on cement-based materials.

## 2. Materials and Methods

### 2.1. Raw Materials

OPC 42.5 was produced by Jidong Cement Company, Xi’an, China, and the LBD was supplied by Pengyuan New Material Company, Xi’an, Shaanxi province, China. The chemical composition of OPC and LBD were summarized in Table 1 and the main mineral components of LBD and OPC can be clearly observed in Figure 1. It should be noted that the technical properties of OPC 42.5 used in this study (provided by the producer) are as follows: specific surface area (Blaine) = 360 m^2^/kg, density = 3.02 g/cm^3^, initial setting time = 2.8 h, and final setting time = 4.7 h. Three types of LBD were considered according to its fineness levels: blaine specific surface area of LBD was 411 m^2^/kg, 623 m^2^/kg and 812 m^2^/kg and noted as A-LBD, B-LBD, and C-LBD, respectively.

### 2.2. Experimental Design

The amount of LBD with different dosages mixed with OPC, and the dosages of LBD were 0, 0.5 wt%, 1.5 wt%, 2.5 wt%, and 4 wt%, respectively (LBD is equivalent to replace the OPC). The LBD/OPC pastes were prepared by manually mixing according to ASTM C305-14, typically 1000 g of solids with 450 g of clear water, and the water-to-solid weight ratio (w/s) of samples used for strength measurement was set as 0.45. The fresh LBD/OPC pastes were poured into molds (40 × 40 × 160 mm) and then they were placed in a curing chamber with a temperature of 20 ± 2 °C and relative humidity about 95% for 3, 7 and 28 d, respectively. The LBD/OPC samples were summarized in Table 2. When the specified curing age was reached, the samples were taken out from the curing chamber, and the molds were removed. The mechanical properties were evaluated by flexural strength test and compressive strength test. Three samples were prepared from each batch for flexural strength test, and six samples were prepared from each batch for compressive strength test. After curing 3, 7, and 28 d from the preparation, roughly 10 g of pastes were withdraw and the hydration reaction was stopped by solvent exchange (add to isopropanol). After two weeks, the isopropanol was drained and then placed in an oven at 40 °C for 7 d, until all of the water was evaporated. The samples were ground using an agate mortar to through 0.075 mm sieve completely, finally the prepared samples were put into a sealed bottle. The preparation process and samples were shown in Figure 2.

### 2.3. Test Methods

#### 2.3.1. Preparing LBD/OPC Pastes

The preparation process of LBD/OPC pastes were as follows: first, the cement mixer should be wiped clean, and then the OPC was replaced by LBD with different dosages and different fineness levels, respectively. The LBD and OPC powder were poured into the mixer, meanwhile 450 g of clear water were slowly added to the mixer. Turn on the machine, the cement mixer was operated at a low speed of 140 rpm for 120 s, stop mixing for 15 s, and then continued to mix at a fast speed of 285 rpm for another 120 s. After mixing was completed, the mixer was stopped and both fresh pastes were poured and compacted into the triple molds (40 × 40 × 160 mm).

#### 2.3.2. Flexural Strength Test

A three-point bending test (flexural) was conducted on the specimens. The flexural strength test standard (T0506 in JTG E30-2005) were used as a reference to determine the testing procedure. The specimen was placed on two supporting points with a certain distance, and a downward load was applied to the upper specimen of the midpoint of the two supporting points. When the two contact points of the specimen form two equal moments, the specimen will break at the midpoint, the flexural strength data were automatically recorded by three-point bending test machine. The loading rate = 50 ± 0.2 N/S, three samples were tested per batch and the results were averaged to ensure the accuracy of the test.

#### 2.3.3. Compressive Strength Test

The compressive strength test standard (T0506 in JTG E30-2005) was used as a reference to determine the testing procedure. The specimen was placed on the bottom plate of the testing machine, and the supporting surface of the specimen should be perpendicular to the top surface. The center of the specimen should be aligned with the center of the bottom plate of the testing machine, and the load should be applied uniformly and continuously until the specimen was crushed. The loading rate = 2.4 ± 0.2 KN/S, six samples were tested per batch and the results were averaged.

#### 2.3.4. X-ray Diffraction (XRD)

XRD (Bruker, Karlsruhe, Germany, D8 Advanced, Cu-Ka) was employed to qualitatively analyze the mineral composition of hydration products. Samples were scanned between 5° to 65° (2θ) and with an integrated step scan of 0.05°/step (2θ). The XRD test was used to determine the mineral components involved in the hydration reaction and resulting composition.

#### 2.3.5. Thermogravimetric and Differential Scanning Calorimetry (TG/DSC)

TG/DSC (TA, SDT Q600 V8.0 Build 95, Nitrogen condition) was used to determine the variation of weight and heat quantity of LBD/OPC pastes with the temperature rise. The nitrogen was used as the shielding gas and the heating temperature ranges between 20 °C and 900 °C with a heating rate of 10 °C/min.

## 3. Results and Analysis

### 3.1. Mechanical Properties

#### 3.1.1. Flexural Strength

The flexural strength of LBD/OPC samples under different fineness levels and dosages were studied systematically. As shown in Figure 3a, the strength of A-LBD/OPC samples reduced first and then increased with increasing of dosages of LBD at all ages. When the dosage of LBD was 0.5%, the strength of the sample was smallest, even lower than that of the control sample. The strength of the samples increased significantly when the dosages of LBD exceeded 0.5%, while the strength of the sample reached the maximum value at the dosage of LBD was 4%. There was no significant difference in the flexural strength between the 3rd and the 7th day. However, there was a significant increase in the strength of 28th day, suggesting that the addition of LBD to OPC had little effect on the early flexural strength, mainly for improve the post strength of the system.

Figure 3b,c present the flexural strength of the sample higher than reference sample (0% additives) at all ages, and the flexural strength increased first and then decreased slightly with the increasing LBD dosages. When the fineness levels of LBD were different, the optimal dosages of LBD were different. For the B-LBD/OPC sample, when the dosage was 2.5%, the strength reached the peak, and the strength of sample was 45.04%, 29.26%, and 24.06% higher than that of reference sample on the 3rd, 7th and 28th days, respectively. For the C-LBD/OPC sample, the optimal dosage was 1.5%, and the flexural strength was improved significantly on the 7th day, indicating that the addition of C-LBD has an impact on the early flexural strength of OPC.

Figure 4 shows the flexural strength contribution rate of the different dosages and fineness levels of LBD to cement-based materials. As shown in Figure 4, the strength contribution rate of LBD/OPC samples was greater than 0, which indicates that LBD has a certain hydration activity to OPC, and has a positive effect for the flexural strength of cement-based materials. From Figure 4a,c, the flexural strength contribution rate reached the maximum when the dosage of B-LBD was 2.5%, the strength contribution rate of the sample increased by about 14% after curing for 3 d. However, for C-LBD, the optimal dosage was 1.5%, and the strength contribution rate of sample increased by 22% after curing for 7 d. From Figure 4c, the strength contribution rate of the C-LBD/OPC samples was generally larger, and its maximum contribution rate can reach 13.5%, which was about 44% higher than the maximum contribution rate of the B-LBD/OPC samples. This indicates that a small amount of C-LBD can make a greater contribution to the flexural strength of cement-based materials due to its relatively larger specific surface area.

#### 3.1.2. Compressive Strength

Figure 5 shows the development law diagram of the compressive strength of the LBD/OPC samples under standard curing conditions. Comparing Figure 3 and Figure 5, it can be seen that the compressive strength and flexural strength of the samples varied with the similar regulation, the fineness levels and the dosages of LBD have an effect on the strength of the samples. However, there were also some small differences between them. Figure 5a shows the compressive strength of A-LBD/OPC samples at ages of 3rd, 7th and 28th days. The experimental results indicate that with the increase of dosages of A-LBD, the strength of the samples increased first and then decreased slightly. When the dosage of LBD was 1.5%, the compressive strength of the sample was reached on peak, at this time, the compressive strength of LBD-OPC samples at the 3, 7, and 28 days was increased by 4.68%, 8.11% and 8.70%, respectively, compared with the reference samples. When the dosage of LBD was 0.5%, the strength of the sample was smallest, even lower than that of the reference sample, the strength of samples at the 3, 7, and 28 days was decreased by 2.5%, 5.08%, and 3.07%, respectively. It is speculated that the reason may be that the specific surface area of A-LBD is relatively small and its hydration activity is low.

Figure 5b,c present that the compressive strength of samples increased first and then decreased slightly with increasing of LBD dosages, and they were larger than the reference sample. The compressive strength of B-LBD/OPC samples reached the peak when the dosage was 1.5%, while for C-LBD/OPC sample reached the maximum value when the dosage was 0.5%. It indicates that the fineness levels of LBD have different effects on the optimal dosage of the compressive strength, and the early and late strength development of samples was also affected by the fineness levels of LBD.

Figure 6 shows the compressive strength contribution rate of the different dosages and fineness levels of LBD to cement-based materials. The strength contribution rate of cement-based materials was greater than 0 after adding LBD powder, besides A-LBD/OPC samples at dosage of 0.5%. Which indicates that LBD has a certain hydration activity to OPC, and it can improve the compressive strength of cement-based materials. From Figure 6a,b, the compressive strength contribution rate reached the maximum when the dosage of B-LBD and C-LBD was 0.5% and 1.5%, respectively. The strength contribution rate of the samples increased by about 7.5%, and 20% for B-LBD and C-LBD after curing for 3 d. From Figure 6c, the strength contribution rate of the A-LBD/OPC sample was generally smaller, it mainly due to the relatively smaller specific surface area.

Combined flexural strength and compressive strength test results, C-LBD powder with dosage 0.5–1.5 wt% is recommended for cement-based material in this study to improve its strength, besides, B-LBD powder with dosage 1.5–2.5 wt% is also a good choice for improving the strength of cement materials.

### 3.2. Hydration Properties

#### 3.2.1. XRD

Figure 7a shows the XRD analysis of LBD/OPC samples under different curing times (A-LBD, with 1.5% dosage). There were diffraction peaks of hydration products appearing such as Ca(OH)_2_, CaCO_3_, MgO, Mg(OH)_2_ and quartz. In addition, the diffraction peaks of secondary minerals such as AFt and C_3_A·CaCO_3_·11H_2_O, as well as the diffraction peaks of C_3_S and C_2_S that are not involved in the hydration reaction can also be found. The diffraction peaks of Ca(OH)_2_ was detected at about 2θ = 18°, 34° and 47°, and the diffraction peaks of Ca(OH)_2_ decreased gradually with the increase of curing time, it shows that more cement clinker participated in the hydration, which made the hydration more thorough. Meanwhile the diffraction peaks development law of C_3_S and C_2_S were similar to the Ca(OH)_2_. MgO and CaCO_3_ are main components of LBD, the diffraction peaks of MgO and CaCO_3_ were detected at about 2θ = 43° and 29°, respectively. When curing for three days, the diffraction peaks of Mg(OH)_2_ was obtained at 2θ = 41°, which indicates that once the MgO is mixed with water, the Mg(OH)_2_ can be formed in a short time.

The XRD of different fineness levels of LBD/OPC samples (7th day with 1.5% dosage) were arranged in Figure 7b. The diffraction peaks of Ca(OH)_2_, C_3_S and C_2_S decreased gradually with the increasing of fineness levels. However the diffraction peaks development law of MgO was opposite compared with Ca(OH)_2_, the main reason is that with the increase of LBD fineness levels, the specific surface area increases accordingly, and the contact area between LBD and OPC increases, and finally more hydration reactions is involved. It should be noted that the diffraction peaks of AFt decreased with increasing of LBD fineness levels, while diffraction peaks of C_3_A·CaCO_3_·11H_2_O show an increasing trend, this means that during the hydration reaction process, AFt was partially converted into AFm, while CaCO_3_ could react with C_3_A to form C_3_A·CaCO_3_·11H_2_O.

Figure 7c shows the XRD results of LBD/OPC samples under different dosages (fineness level: C-LBD, curing time: 7th day). It can be seen that the dosages of MgO and CaCO_3_ increased with the increasing of LBD dosages.

#### 3.2.2. TG/DSC

The thermogravimetric results of LBD/OPC samples were shown in Figure 8 (7th day, C-LBD). The TG curve could quantitatively provide the mass loss of each individual phase. Three obvious mass losses could be observed from Figure 8a: one from 100 °C to about 300 °C with a larger span, the second in the 400–500 °C range; and the last from 600 °C to 700 °C. According to the related works in the literatures [7,24], the relative mass loss in the first weightlessness period was about 6.1%, which is mainly caused by the physical dehydration of hydration products. The physical dehydration mainly included the following parts: (i) the dehydration temperature of C-S-H gel was about 105 °C which produced by cement hydration; (ii) the dehydration temperature of AFt was about 125—140 °C; (iii) the dehydration temperature of AFm was about 175° C; (iv) and the dehydration temperature of C_3_A·CaCO_3_·11H_2_O was about 200—300 °C. The second loss was ascribed to the decomposition of Ca(OH)_2_ with a relative mass of 4.5%. The third relative mass loss was smallest, only 2.7%, and the decomposition CaCO_3_ or CaMg(CO_3_)_2_ to produce CO_2_ was the main reason for the mass loss. This shows that the mass loss decreased first and then increased significantly with increasing LBD dosages. When the LBD dosage was 0.5%, the mass loss was the smallest, even lower than the reference sample. However, the mass loss was largest when the LBD dosage was 4%, which is due to the dosage of CaCO_3_ increases with increasing of LBD, and more CO_2_ is produced from the decomposition of CaCO_3_.

Figure 8b shows the DSC results of samples at different dosages (7th day, C-LBD). It can be seen that the endothermic events occurred from 100 to 200 °C and 400—500 °C, while small endothermic events occurred around 650—750 °C. The first is mainly due to the physical dehydration of AFt and AFm; the second corresponds to Ca(OH)_2_ decomposition to absorb heat; the last one is attributable to CaCO_3_ or CaMg(CO_3_)_2_ decomposition producing CO_2_.

The state of the water in the LBD/OPC samples was very complex, and it was difficult to strictly distinguish the physical combination or chemical combination of the water. For research convenience, dehydration below 105 °C is generally considered to be evaporable water and the dehydration at 105—500 °C is identified as non-evaporable water [25,26,27]. Table 3 shows the mass losses of C-LBD/OPC samples at different dosages (7th day, C-LBD). It was found that the mass losses percentage of the hydration products decreased first and then increased with the increase of C-LBD dosages at all of the different temperature ranges. When the LBD dosage was 1.5%, the mass losses was minimized, the loss rate of evaporable water, non-evaporable water and decarbonation of CaCO_3_ or CaMg(CO_3_)_2_ in the samples were 3.16%, 10.99%, and 2.89%, respectively. The results also indicated that the mass loss rate of evaporable water of the samples was lower than reference sample when added to C-LBD, this means that C-LBD has a positive effect on cement hydration. The loss rate of non-evaporable water and decarbonation of CaCO_3_ or CaMg(CO_3_)_2_ in the pastes were 11.41% and 3.63%, respectively. When the LBD dosage was 4%, the mass losses were larger than reference sample, mainly since when the fineness of LBD increases to a certain level, it has an adverse effect on the hydration of the LBD/OPC paste.

### 3.3. The Reaction Degree of LBD

To describe the hydration process of LBD/OPC samples, the reactivity of LBD was determined by calculating the difference between the initial CaCO_3_ dosage and the final CaCO_3_ dosage, the initial CaCO_3_ dosage was determined as Equation (1), and the mass loss at approximately 650—750 °C was used to measure the final CaCO_3_ dosage as percentage by weight (wt%) using stoichiometry as presented in Equation (2) [28,29]
(1)$$M=\frac{M_{1}\;}{M_{2}+M_{3}+M_{4}}\times 100\%$$
(2)$$T=M_{5}\cdot \frac{M_{6}{\mathrm{orM}}_{7}\;}{M_{8}\mathrm{or}2M_{8}}$$
R = M − T(3)
where M is the initial CaCO_3_ dosage; T is the final CaCO_3_ dosage; R is the reaction degree of LBD; M_1_ is the mass of LBD (blended and interground), M_2_ is the mass of OPC, M_3_ is the mass of LBD, M_4_ is the mass of water in the pastes, M_5_ is the mass loss of pastes at 650—750 °C, M_6_ is the mass of CaCO_3_, M_7_ is the mass of CaMg(CO_3_)_2_, M_8_ is the mass of CO_2_.

According to the results of TG curve, the reaction degree of LBD in OPC can be calculated by Equations (1)—(3) and shown in Figure 9 (7th day, C-LBD). Obviously, the dosages of C-LBD has a bigger effect on the degree of hydration reactivity, which is consistent with the conclusions of the literature [30]. When the dosage of LBD was 1.5%, the reaction degree was maximized, and the reaction degree is about 4.8%. This is consistent with the flexural strength results for C-LBD/OPC samples.

## 4. Discussion

According to the data provided by flexural strength and compressive strength, the fineness levels and the dosages of LBD have an influence on the strength of the cement-based materials, this remarkable phenomenon can be observed at all curves. Typically, in a certain fineness levels range, the finer the LBD, the larger specific surface area, the greater contact area with the cement, and the more hydration reactions involved. In the early stages of cement hydration, the main hydration products were AFt, Ca(OH)_2_ and C_2_S, C_3_S; with increasing of the curing time, part of the AFt translated into AFm. The main components of LBD are MgO and CaCO_3_. Some studies have pointed out that CaCO_3_ can react with C_3_A to produce C_3_A·CaCO_3_·11H_2_O, which prevents the conversion of AFt to AFm [24,25,26]. At the same time, the formation of C_3_A·CaCO_3_·11H_2_O could improve the pore structure of pastes and enhance the strength of the LBD/OPC samples [27,28]. However, when the fineness levels of LBD increased to a certain extent, the specific surface area was too large, and the agglomeration phenomenon was prone to occur. With the increase of LBD dosages, the aggregation phenomenon will affect the contact area between LBD and cement powder, thereby reducing the amount of LBD participating in the cement hydration reaction and reducing the strength of the paste. Therefore, the excessive LBD would have detrimental effect on the strength of the LBD/OPC pastes when it has sufficient fineness.

The presented results obtained by XRD, TG, DSC and reaction degree on LBD/OPC pastes hydration showed that the carbonated aluminate phase was formed. This new phase could improve the microstructure of the sample and make it more compact, which plays an important role in improving the strength of the LBD/OPC pastes [31]. Besides, LBD has physical effects such as nucleation, dilution and filling in OPC, the synergistic effect of the carbonated aluminate phase and this physical effect could improve the properties of OPC. The filling effect of the LBD is related to its fineness and particle size distribution, the dilution effect is related to the dosages of LBD, and the dilution effect mainly occurs in the early stage of cement hydration [32,33,34]. Some studies have pointed out that the carbonated aluminate phase was formed according to the following Equations (4)—(8) [35]. However, professor Zajac [36] held a different view on the formation of the carbonated aluminate; he believes that the formation of the carbonated aluminate was indirect (that is to say, the formation of the carbonated aluminate phase is divided into two steps: (i) the LBD was dissolved in the water; (ii) the CaCO_3_ was reacted with aluminum-containing components in the pastes to form carbonated aluminate furtherly). At the same time, professor Zajac pointed out that Mg(OH)_2_ is unstable and it can react with aluminum containing components in the pastes to form hydrotalcite. However, in this study, only the diffraction peak of Mg(OH)_2_ were found in LBD/OPC pastes under different LBD dosages and curing time by XRD experiments, and no diffraction peak of hydrotalcite was found, it is different from professor Zajac idea. However, regardless of whether it is an indirect reaction mechanism or a direct reaction mechanism, the reaction occurs after the LBD was dissolved. When the fineness levels of LBD were appropriately increased, the dissolution rate was accelerated, and the reaction rate and degree were correspondingly improved.

LBD could release CO_3_^2−^ ions and then combine with alumina to form a carboaluminates phase, which is mainly based on the dissolution rate of LBD. Some studies have shown that LBD has a lower dissolution [37,38,39,40], which indicates that the LBD reacts slowly in OPC system, while carboaluminates increased with curing time, which prevented the conversion of AFt to AFm. AFt can be combined with carboaluminates to ensure better filling of pore structure and well refined pore structure, and increasing the strength of Portland cement.
CaMg(CO_3_)_2_ → CaCO_3_ + MgCO_3_(4)
Mg(CO_3_)_2_ → MgO + CO_2_(5)
MgO + H_2_O → Mg(OH)_2_(6)
CaMg(CO_3_)_2_ + Ca (OH)_2_ → CaCO_3_ + Mg(OH)_2_(7)
CaCO_3_ + C_3_A + 11H_2_O → C_3_A·CaCO_3_·11H_2_O(8)

Limestone is the primary raw material for the production of clinker in the cement production process and has a positive effect on the strength and heat of hydration of Portland—limestone cement. However, dolomite and LBD are not to be used as raw materials for clinker production, as they are a potential carbonate source and could be utilized together with Portland cement. Compared with limestone mixed with cement, there was no negative effect after LBD mixed with cement. The performance of dolomite mixed with cement is comparable to that of limestone mixed with cement, and no negative effect is found, which is related to the clinker produced by limestone containing dolomite [14].

## 5. Conclusions

Based on the experimental results in this study, the mechanical properties and hydration properties of LBD/OPC sample with different LBD dosages (0, 0.5, 1.5, 2.5, 4.0 wt%) and fineness levels (A, B and C) were investigated at the 3rd, 7th, and 28th days. the following conclusions can be drawn:

(1)LBD powder has a certain hydration activity to OPC materials, and it can improve the post strength of the paste significantly. The flexural strength of LBD/OPC samples was higher than reference sample at all ages. For the B-LBD/OPC sample, the flexural strength contribution rate reached the maximum with dosage of 2.5%, and for the C-LBD/OPC sample, the optimal dosage was 1.5%.(2)The compressive strength of LBD/OPC samples increased first and then decreased slightly with increasing of LBD dosages. The compressive strength reached its peak when the B-LBD dosage was 1.5%, and C-LBD dosage was 0.5%, and the strength contribution rate of the samples increased by about 7.5%, and 20% for B-LBD and C-LBD samples after curing for 3 days. In conclusion, C-LBD powder with dosages 0.5–1.5 wt% and B-LBD powder with dosages 1.5–2.5 wt% are recommended for cement-based material.(3)The main components of LBD are MgO and CaCO_3_, of which MgO could react with water rapidly to form Mg(OH)_2_, and CaCO_3_ could react with C_3_A to produce C_3_A·CaCO_3_·11H_2_O, which prevents the conversion of AFt to AFm.(4)Three principle mass losses can be observed: one from 100 °C to about 300 °C with a larger span, the second approximately in the 400—500 °C range, and the last from 600 °C to 700 °C. Three endothermic events were found in the DSC curve, which were caused by the physical dehydration of AFt and AFm, Ca(OH)_2_ decomposition to absorb heat and CaCO_3_ or CaMg(CO_3_)_2_ decomposition to produce CO_2_, respectively.

## Figures and Tables

**Figure 1 materials-15-05798-f001:**
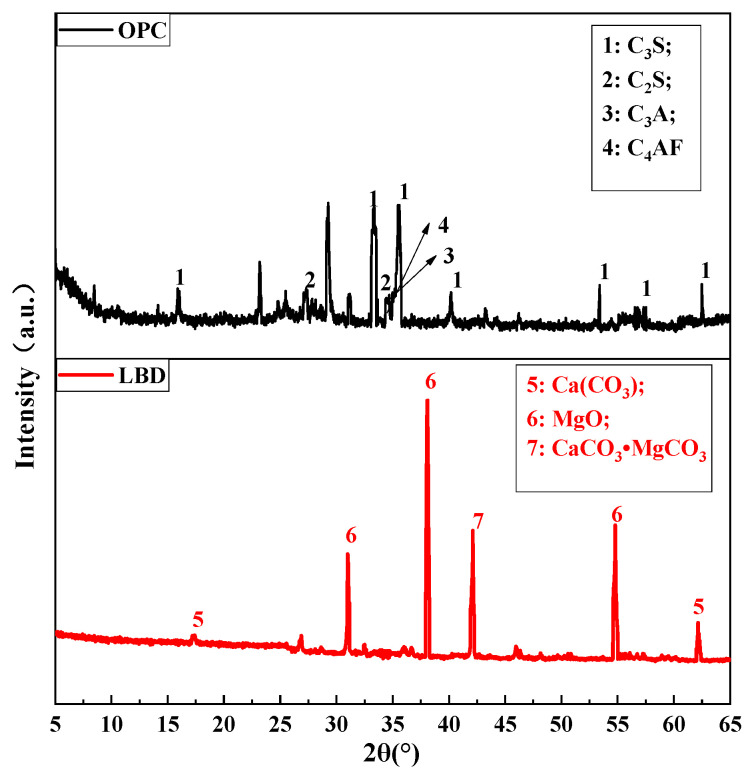
XRD patterns for OPC and LBD powders.

**Figure 2 materials-15-05798-f002:**
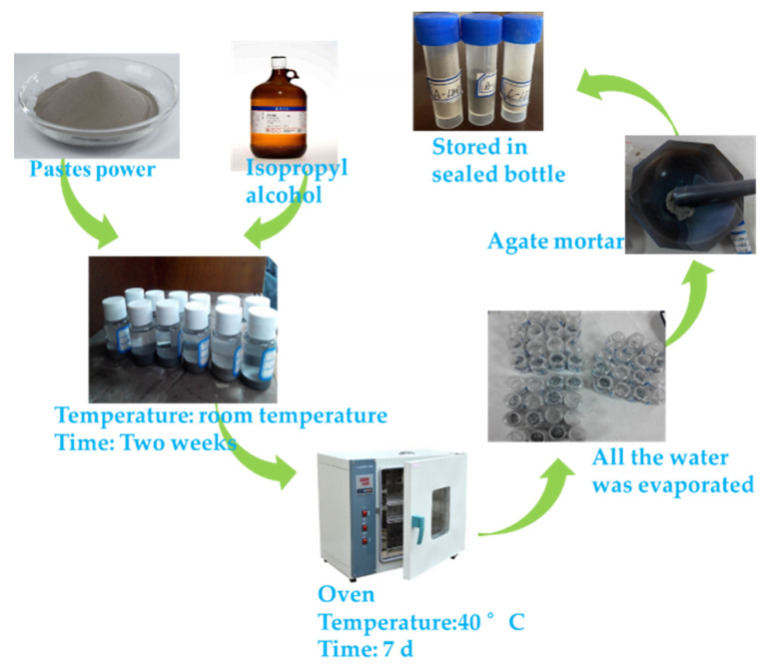
The operation process of samples.

**Figure 3 materials-15-05798-f003:**
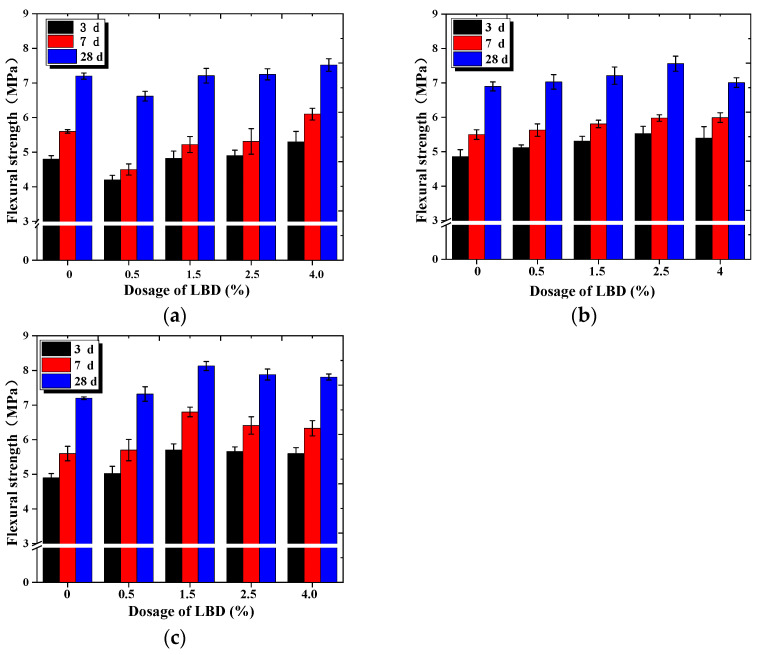
Flexural strength of LBD/OPC samples: (**a**) A-LBD/OPC samples; (**b**) B-LBD/OPC samples; (**c**) C-LBD/OPC samples.

**Figure 4 materials-15-05798-f004:**
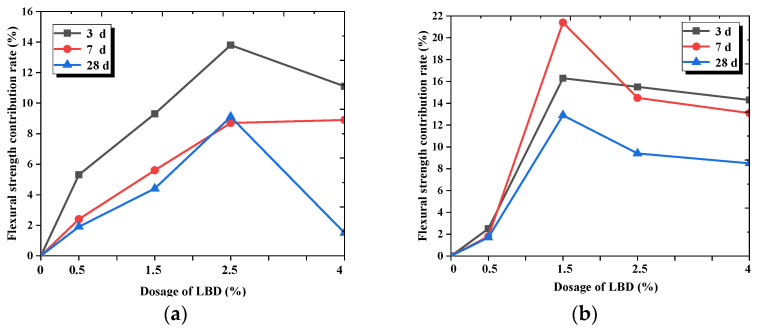
Flexural strength contribution rate of LBD to OPC materials: (**a**) Influence of B-LBD dosages; (**b**) influence of C-LBD dosages; (**c**) influence of LBD fineness levels (curing for 28 d).

**Figure 5 materials-15-05798-f005:**
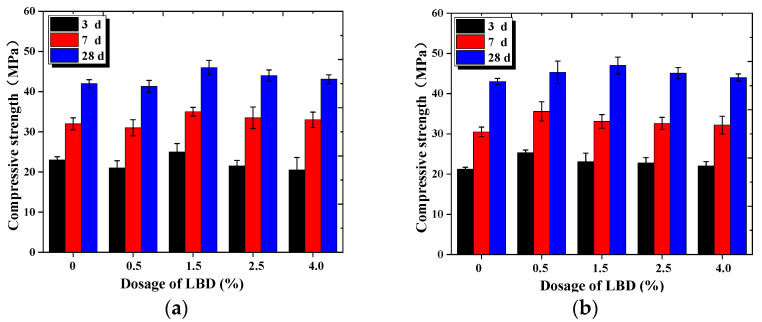
Compressive strength of LBD/OPC samples: (**a**) A-LBD/OPC samples; (**b**) B-LBD/OPC samples; (**c**) C-LBD/OPC samples.

**Figure 6 materials-15-05798-f006:**
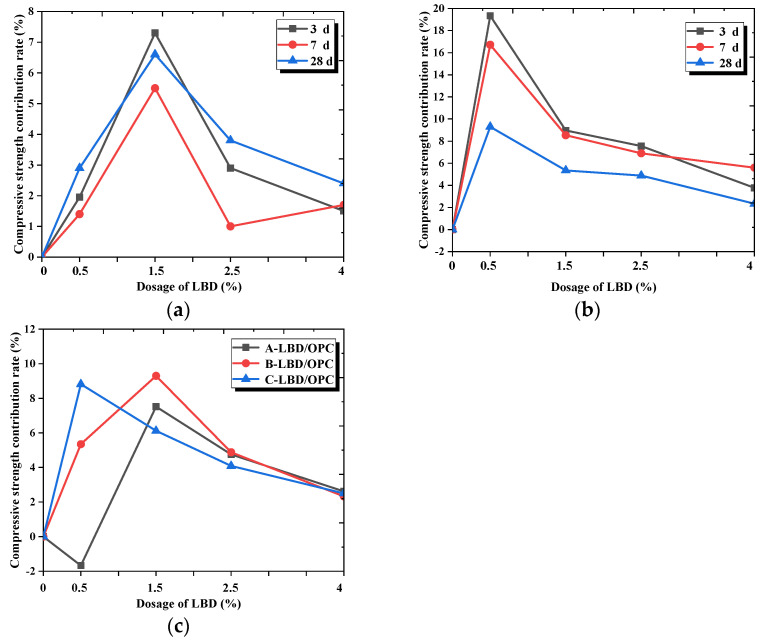
Compressive strength contribution rate of LBD to OPC materials: (**a**) Influence of B-LBD dosages; (**b**) Influence of C-LBD dosages; (**c**) Influence of LBD fineness levels (curing for 28 d).

**Figure 7 materials-15-05798-f007:**
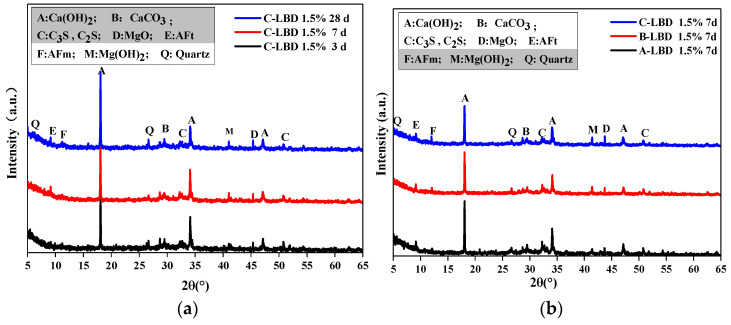
XRD diffractograms LBD/OPC pastes: (**a**) Influence of time; (**b**) influence of fineness; (**c**) influence of dosage.

**Figure 8 materials-15-05798-f008:**
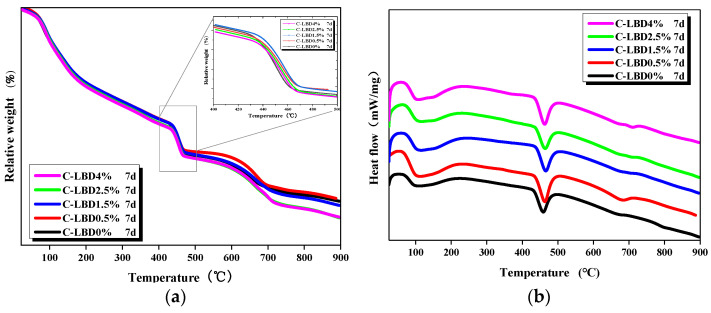
Thermal analysis curves of LBD/OPC pastes: (**a**) TG; (**b**) DSC.

**Figure 9 materials-15-05798-f009:**
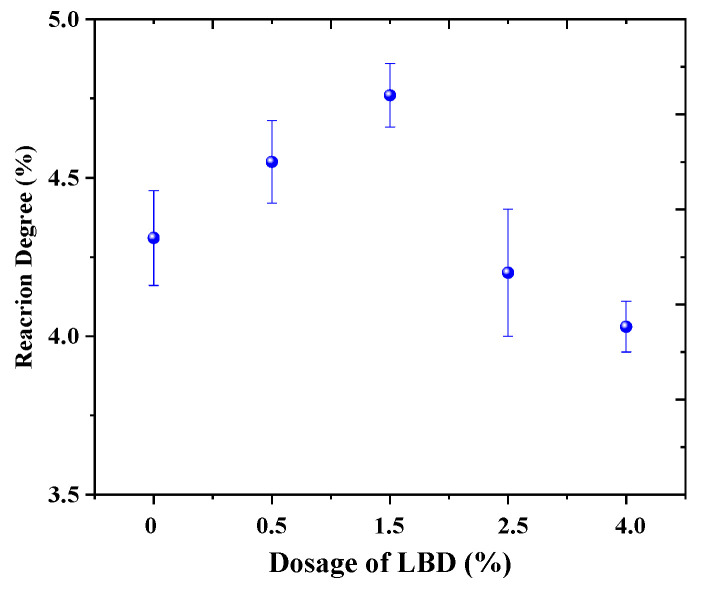
Reaction degree of LBD/OPC pastes.

**Table 1 materials-15-05798-t001:** Chemical composition of raw materials.

Raw Materials	CaO	SiO_2_	Al_2_O_3_	Fe_2_O_3_	SO_3_	MgO	K_2_O	LOI
OPC 42.5	65.8	18.00	5.2	4.8	3.00	1.3	1.1	0.8
LBD	32.4	0.74	0.17	0.12	0.09	21.3	0.01	45.17

**Table 2 materials-15-05798-t002:** Test samples of LBD/OPC pastes.

Marked Number	Dosages of LBD (%)	Time (Days)	W/S Ratio	Temperature and Humidity
A-LBD/OPC	0, 0.5, 1.5, 2.5, 4	3, 7, 28	0.45	20 °C, 95%
B-LBD/OPC				
C-LBD/OPC				

Where: A-LBD/OPC noted as fineness levels of LBD was A equivalent to replace the OPC; B-LBD/OPC noted as fineness levels of LBD was B equivalent to replace the OPC; C-LBD/OPC noted as fineness levels of LBD was C equivalent to replace the OPC.

**Table 3 materials-15-05798-t003:** Mass losses of the paste at different temperature ranges and corresponding processes.

Dosages of C-LBD	Evaporable Water (below 105 °C), %	Non-Evaporable Water (105–500 °C), %	Decarbonation of CaCO3 or CaMg(CO3)2, %
0%	3.525	11.17	3.24
0.5%	3.36	11.04	3.17
1.5%	3.16	10.99	2.89
2.5%	3.2	11.24	3.45
4%	3.4	11.41	3.63

## Data Availability

Not applicable.

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
