# Peer review of "Influence of Fineness Levels and Dosages of Light-Burned Dolomite on Portland Cement Performance"

_materials, 2022, doi:10.3390/ma15165798_

Round 1

Reviewer 1 Report

This paper extensively compiled and argued the results with comparing the strength, characterization studies.

Ø  The introduction part is not explicitly written up to the scientific required.

Ø  There is need of strong literature support with justification of novelty of work.

Ø  Materials physical and chemical properties are not clearly discussed.

Ø  The research is not specific applied to any particular applications.

Ø   Experimental test setup needs more clarity with photos may give much more understanding.

Ø  References are very limited.

Ø  The recent literature must cite and discussed.

Reviewer 2 Report

The research in this paper is novel and has important reference value for promoting its engineering applications. This work could be a valid contribution to the research community, but some improvements should be made in order to increase the quality of the paper. The reviewer suggests to the authors to take into account the following comments:

1)      The introductory part needs to be improved by providing a more comprehensive review for the recently work

2)      The differences from previous studies must be specified clearly in introduction section.

3)      The novelty of this work must be more demonstrated.

4)      How many samples were repeated for each test? Standard deviations should be provided.

5)      Conclusion section is inadequate and it should be significantly improved.

6)      The conclusion part is limited, and it is only a summary of the results. We need technical and general advices here which can be used by others (both researches and engineers)

7)      The text severely suffers from multiple grammatical errors as well as awkward vocabulary usage. Besides, there are many mistakes in case of sentence structure, use of articles (the, a, etc.), and verb tenses.

Reviewer 3 Report

The paper presents “Influence of Fineness and Dosage of Light-Burned Dolomite on Portland Cement Performance”; this paper studies the influence of fineness levels, dosage, and curing times of LBD/OPC pastes. The hydration properties (using XRD, TGA, and DSC) and mechanics performances were investigated. The paper was well written. The following comments must be taken to improve the manuscript:

1-      Figure 1, XRD not XPD.

2-      The subtitle 2.2 & 2.3 “Experimental design” was repeated.

3-      In line 120, the authors stated in the LBD/OPC mix, “then continues to mix at a fast speed of 285 rpm for another 120 min”. Why mixed for 120 min.? I think you mean 120s.

4-      Figure 6 stated in the text before Figs. 4 & 5. Rearrangement of the figures according to priority.

5-      The effect of fineness of LBD on the compressive strength should be included in the conclusion section.

Reviewer 4 Report

The authors tried to present experimental findings on the effect of light-burned dolomite powders (LBD) on Portland cement and evaluate the influence of LBD dosage and fineness levels on the mechanical properties and hydration properties of Portland cement by XRD and TGA tests. The subject of this manuscript is within the scope of this journal. Although there are problems, it can be recommended for publication if the following suggestions and comments will be considered and responded:

The abstract does not include the background knowledge of this study. Furthermore, this section lacks the critical values from the results and the novelty of this work. Most significant numerical values should be noted in the abstract section. For example, the authors can provide the optimum content of LBD on the mechanical properties of specimens. Meanwhile, they can provide the effect of varying the finesses of LBD on the mechanical and microstructure of specimens.

The introduction is unable to give the reader detailed background and possible wide application of this study. This section needs to be more emphasized the research work on the effect of various content of light-burned dolomite powders (LBD) on the mechanical and microstructural properties of concrete. There are substantial previous works in these fields.

The authors should clarify the reason why they choose dosages of LBD 0, 0.5 %, 1.5 %, 2.5 %, and 4 %. Furthermore, a detailed explanation of the past, present, research gaps, and future scope of this study. Most of the references are too old (before 2018), so the authors must add recent literature.

How did the authors prepare and select the samples for the XRD and TGA tests? The samples passed from which sieve? The sieve size play a great role in the accuracy of the results.

The authors can remove figure 3. This figure does not add any scientific content to the manuscript.

Error bars are missing from all bar charts, and it is unclear how many replicates are measured for each experiment.

The author provided pieces of evidence from previous microstructural results to explain the effect of varying the content and the finesses of LBD on the flexural and compressive strength, and these pieces of evidence are not detailed enough. They must implement their findings (the X-ray and TGA test) to justify their results.

In figure 4 and 5, provide a correct caption for figures 4a-c.

The results showed that when the dosage of LBD was 0.5 %, the strength of the sample was lower than normal specimens. However, the authors couldn’t justify this loss of strength. Are the authors sure that the test was conducted correctly, and that this data is not the reason for the wrong test?

Reviewer 5 Report

The approach to presenting research data in this manuscript is quite weak and requires major improvement. The manuscript needs language editing. The author may take professional help in this regard.

1. Please highlight the key findings in the abstract and no need to provide all study results in the abstract.

2. Introduction portion is to weak, need some serious improvement.

3. Please improve the resolution. Barely see the text.

4. The Experiment program is lacking in novelty, there is no significant contribution by this manuscript in existing literature.

5. Please change the color schemes for the graphs.

6. Kindly add statistical analysis for graphs presented.

7. The reviewer hoped to see the results of a additional testing.

8. Please provide more in-depth chemical discussion.

9. Kindly provide the actual raw file as supplementary file, and also mention the individual peaks. The results presented in manuscript are extensively smoothies.

10. kindly support your results with previous literature.

11. Has the XRD instrument been calibrated to obtain the quantitative results.

12. It is important to provide SEM and Spot EDS results. kindly add them as well.

Round 2

Reviewer 2 Report

The authors made corrections based on the reviewers comments.

The paper can be accepted.

Author Response

The authors are grateful to the reviewers of this paper for their comments and suggestions.

Reviewer 4 Report

The authors responded to the questions and made the necessary corrections. The manuscript is ready for publication in the journal of Materials.

Author Response

(The authors gave the same response as above.)
